# Improving the Long-Term Mechanical Properties of Thermoplastic Short Natural Fiber Compounds by Using Alternative Matrices

**DOI:** 10.3390/biomimetics10010046

**Published:** 2025-01-13

**Authors:** Renato Lemos Cosse, Tobias van der Most, Vincent S. D. Voet, Rudy Folkersma, Katja Loos

**Affiliations:** 1Circular Plastics, Academy Tech & Design, NHL Stenden University of Applied Sciences, Van Schaikweg 94, 7811 KL Emmen, The Netherlands; r.lemos.cosse@rug.nl (R.L.C.); tobias.van.der.most@nhlstenden.com (T.v.d.M.); vincent.voet@nhlstenden.com (V.S.D.V.); 2Macromolecular Chemistry and New Polymeric Materials, Zernike Institute for Advanced Materials, University of Groningen, Nijenborgh 3, 9747 AG Groningen, The Netherlands

**Keywords:** wood plastic composites, natural fiber reinforcement, thermoplastic matrices, mechanical properties in hostile environment, water absorption, mechanical properties under moisture, creep, heat deflection temperature, free–thaw cycle testing

## Abstract

Wood plastic composites (WPCs) offer a means to reduce the carbon footprint by incorporating natural fibers to enhance the mechanical properties. However, there is limited information on the mechanical properties of these materials under hostile conditions. This study evaluated composites of polypropylene (PP), polystyrene (PS), and polylactic acid (PLA) processed via extrusion and injection molding. Tests were conducted on tensile and flexural strength and modulus, heat deflection temperature (HDT), and creep analysis under varying relative humidity conditions (10% and 90%) and water immersion, followed by freeze—thaw cycles. The addition of fibers generally improved the mechanical properties but increased water absorption. HDT and creep were dependent on the crystallinity of the composites. PLA and PS demonstrated a superior overall performance, except for their impact properties, where PP was slightly better than PLA.

## 1. Introduction

Environmental concerns have awakened society to the consequences of today’s production and consumption regime. The awareness of diminishing the collateral effects of human activities on the planet has pushed the research toward better resource management. Plastics are well-known as fossil-based materials that persist for many years in the environment. Although plastics present many advantages for different applications, the end life of these products has generated challenges in managing material waste [1]. In this direction, numerous approaches exist to decrease the need for fossil-based materials and introduce bio-based materials.

One of the efforts to reduce the carbon footprint of polymers is to synthesize bio-based thermoset resins by epoxidizing many different crops. Soybeans [2,3], cashew nutshell liquid [4,5], sunflower [6], canola [7], corn [8], cottonseed [9], jatropha [10], mahua [11], limonene [12], and tea leaf oils [13] are among those that have succeeded. In addition, many years of research have been undertaken on the use of natural fibers for a variety of purposes, with composite materials being one of the primary areas of study [14]. Biocomposites are an alternative term for natural fiber composites (NFCs). In NFCs, natural fibers partially replace polymers, imparting better properties in the attempt to substitute conventional composites in several applications. Even with the capability of being derived from renewable sources, thermoset resins have the drawback of not being recyclable in their end life, which puts thermoplastics under the spotlight. Unlike thermosets, thermoplastics are not crosslinked and are remoldable, meaning that they undergo mechanical recycling. However, in general, for construction applications, thermoplastic mechanical properties are inferior, making reinforcement necessary [15]. It is important that the reinforcement allows composites to be processed via well-established processing routes, and for this purpose, wood fibers have been explored. When wood fibers are compounded with thermoplastics, they are called wood plastic composites (WPCs).

WPC has become a growing topic over the last two decades. This type of composite material benefits from the high availability of raw materials worldwide and is often obtained from waste from other industrial activities [16,17]. The advent of renewable thermoplastics on an industrial scale has boosted the use of these materials in composites. Currently, it is possible to produce polyolefins, such as polyethylene (PE) [18] and polypropylene (PP) [19], derived from bioethanol, polystyrene from lignin [20], polyesters such as polyhydroxyalkanoates (PHAs) [21] from microorganisms, and polylactic acid (PLA) from starch sources [22]. The latter have attractive mechanical properties and are compostable under industrial conditions. Moreover, PLA presents a processing window compatible with polyolefins, unlike what occurs with PHAs [21]. The sum of these characteristics of PLA makes it possible to use this material in many areas already dominated by polyolefins.

However, WPCs have limited applications for use that require greater mechanical stress, in addition to the fact that creep occurs in the life cycle of these materials [23]. The mechanical performance of WPCs requires much interaction between the polymeric matrix and the fibers inserted during processing [24]. Therefore, several strategies range from the chemical modification of polymers to compatibilizing agents [25,26,27] to the chemical and physical treatment of fibers [14]. These aspects have led to the research on optimizing interaction factors with individual polymer matrices, opening the possibility for more reliable comparisons between different thermoplastic matrices. Although many materials undergo similar processing routes, the parameters used differ, as does the way in which their mechanical characteristics are tested. For instance, the crystallinity of semicrystalline polymers is affected by the processing temperature and cooling rate, thereby affecting their thermal and mechanical performance.

Inspired by biomimetics, this study draws from natural systems, specifically the structural and functional characteristics of wood fibers, to improve the performance of polymer composites [28]. By integrating natural fibers, such as wood into thermoplastic matrices, the research mirrors nature’s strategies for creating lightweight yet strong materials. This biomimetic approach not only enhances mechanical properties, including the tensile and flexural strength, but also aligns with environmental sustainability by reducing the dependence on fossil-based materials [29]. Furthermore, by mimicking the hierarchical structure found in natural composites, this study contributes to advancing materials engineering, particularly for applications in challenging environments [30].

Considering these adverse factors, the present work aims to manufacture wood plastic composites through conventional processing routes (extrusion and injection molding) using three different thermoplastic matrices (PP, PS, and PLA). The effects of better adhesion between the fiber and matrix and WPC annealing on the thermomechanical performance of the WPCs were studied and compared. Thus, tensile, flexural, impact, and heat deflection temperature (HDT) tests were performed under the influence of water absorption. Furthermore, the influence of crystallization and environmental conditions, such as relative humidity and freeze–thaw cycles, on creep behavior was analyzed. In order to compare the properties of WPCs with those of commonly used composites, glass fiber-reinforced PP was manufactured and tested in parallel.

## 2. Materials and Methods

### 2.1. Materials

The chosen matrices satisfy the condition that the processing window temperature remains below 200 °C, the temperature at which natural fibers undergo the most significant degradation. Under this frame, a copolymer polypropylene (PP), Sabic PHC27 (CAS: 9010-79-1), with a melt flow rate (MRF) of 14 g/10 min (230 °C/2.16 kg), was used as a baseline. The preference for a copolymer instead of a homopolymer is the trade-off between the modulus and impact strength at low temperatures. Pure polystyrene BASF 158K (CAS: 9003-53-6), MFR 3.7 g/10 min (200 °C/5 kg), was chosen as the standard. Polylactic acid (PLA) was obtained from NatureWorks, 4043D (PLA1, CAS: 9051-89-2) with 6252D (PLA2, CAS: 26023-30-3) with MFRs of 6 and 70–85 g/10 min at 210 °C and 2.16 kg. According to the PLA supplier, the two variants present different d-isomer contents (PLA 4043D has a higher content than PLA 6252D), which may influence not only the viscosity but also the crystallization rate [31,32].

The following compatibilizers were obtained from Sigma-Aldrich: PP-grafted maleic anhydride (MAPP, CAS: 25722-45-6) and styrene–maleic anhydride (SMA, CAS: 26762-29-8) for modifying polystyrene. Acetone (CAS: 67-64-1), maleic anhydride (MA, CAS: 108-31-6), and dicumyl peroxide (DCP, CAS: 80-43-3) purchased from Sigma-Aldrich were used for the synthesis of PLA-grafted maleic anhydride (PLA-G-MA).

Two different grades of Arbocel (J. Rettenmaier & Söhne GMBH, Rosenberg, German) wood fibers were used as reinforcements, C320 (F) and C400 (C), whose screen analysis is described Appendix A. The glass fiber ChopVantage^®^ HP 3299 (G), CAS: 65997-17-3, PPG Industries, Pittsburgh, PA, USA) was 4.5 mm in length and had an average diameter of 13 µm (see Appendix A for micrographs).

Two different salts were used for the moisture content test: potassium sulfate (K2SO4, CAS: 7778-80-5) and lithium chloride (LiCl, CAS: 7447-41-8). At 23 °C, saturated salt solutions (with salt particles not dissolved in water) should create relative humidity values of 90% and 11% in a closed container.

### 2.2. Methods

#### 2.2.1. Compounding

PLA 4043D and 6252D were dried in a single desiccant tower, Piovan DS506, at 80 °C for 12 h, while PS and PP were dried at 90 °C and 50 °C for two hours. The wood fibers were dried for 4 days at 105 °C in a ventilation oven (VWR Ventiline, Haryana, India), followed by 4 days under vacuum (Thermo Scientific Vacutherm, Waltham, MA, USA) at 105 °C.

The preparation of PLA-G-MA followed the reactive extrusion protocol described by Zhang et al. [33], since previous studies [34,35,36], showed its efficiency in enhancing the mechanical properties of NFCs. Four identical mixtures were made: 5 g of DCP and 15 g of MA were dissolved in 100 mL of acetone, and each mixture was then poured over and mixed with 1 kg of PLA1. Then, the material was left in a vacuum environment for one hour to allow acetone evaporation. Four kilograms of PLA coated with DCP and MA were compounded with the same settings as for the PLA-wood compounds. Appendix A shows the schematic chemical reaction that occurs throughout the compounder.

A nearly identical method was used for the direct compounding of PLA1, MA, DCP, and wood fiber. Two identical mixtures of 3.75 g of DCP and 11.25 g of MA were dissolved in 75 mL of acetone and then poured over 2.5 kg of 4043D for a total of 5 kg. Table 1 contains the list of materials prepared for testing. The material code is as follows: polymer (+additives) (+fiber with fiber content in volume percent). All polymer compounds were tested as their compatibilized versions. The PLA variations were made around PLA1+F35 with one parameter varying at a time (polymer viscosity, fiber load, fiber size, or additive/compatibilizer). Figure 1 graphically shows the difference in the compounding route selected for manufacturing the samples. The preparation methods for the compatibilization of the composites were tested differently to verify whether one processing method would significantly differ from the other in terms of the mechanical properties.

Compounding was carried out in a Kraus-Maffei Berstorff ZE25Ax43D UTX in combination with three Brabender gravimetric feeders (the screw and barrel configuration are depicted in Appendix A).

MAPP and PLA-G-MA were dry-blended with PP and PLA1, respectively, and then placed in a feeder. The PS modifier, powder SMA, was fed with a third feeder.

Both the compounder and side feeder screw speeds were set to 200 rpm. The vacuum port was maintained at 0.7 bar relative negative pressure. The temperature settings for the various materials are presented in Table 2. The throughput was set to be constant by adjusting the output of the polymer (and compatibilizer) feeder relative to the maximum output of the fiber feeder (set to 3 kg/h).

#### 2.2.2. Injection Molding

Before injection molding, all materials were dried under vacuum for 72 h at 65 °C. Injection molding was performed with an Engel E-mac 50 with a 30 mm cylinder and 4 heating zones (Table 3). The mold used to produce the test samples was the Axxicon AIM quick change mold with Inserts ISO A (2 tensile specimens) and ISO B (4 impact/flexural specimens).

The injection speed was set to fill rates of 16 and 32 cm^3^/s for A and B, respectively, resulting in a consistent fill rate of 200 mm/s for the test area of the test samples according to ISO 294 (1996) [37]. The back pressure of the screw was set to 80 bar, the screw speed was set to 0.25 m/s, the dosage was set to 36.5 cm^3^, and decompression was set to 6 or 8 cm^3^ for pure polymers or compounds, respectively. The holding pressure and time, switching pressure, and cooling time were altered to create flat specimens that were still able to eject.

#### 2.2.3. Thermal Annealing of PLA

To increase crystallization, selected PLA1 samples were placed between a set of glass plates and parchment paper. This stack was then placed in a preheated ventilation oven set to 105 °C and left for two hours. Then, the stack was removed from the oven and left to cool to room temperature before removing the samples. From this point forward, samples that underwent thermal annealing will be indicated with the word “oven” in parentheses.

#### 2.2.4. Static Mechanical Test

All static tests were performed after climatizing the test samples at 23 °C and 50% relative humidity (ISO 291 (2008)) [38] for two weeks to allow moisture equilibrium. Static tensile and flexural testing was performed on a Zwick UPM ZMART. PRO 14740.

Tensile testing was performed in accordance with ISO 527-2 (2012) [39] on the aforementioned UPM equipped with a 100 kN load cell, hydraulic vice clamping, and Zwick BZ1-EXZW003 macro extensometer with a resolution of 1 μm. The tensile modulus was measured with a test speed of 1 mm/min with the use of an extensometer for accurate deformation measurements. After the tensile modulus measurement, the extensometer was automatically dismounted, and the speed was increased to 5 mm/min to measure the remainder of the curve. The tensile test for the unfilled PP was stopped after reaching a 50% strain since, from past experience, it is known that this material will not break before the hardware limit is reached (>400%), and at a 50% strain, the material has already yielded.

Flexural testing was performed in accordance with ISO 178 (2010) [40] on the same UPM but equipped with a 5 kN load cell, the adjustable 3-point bending fixture was set to a span of 64 mm, and a Zwick BZ1-EXZW013 3-point bending extensometer with a resolution of 1 μm. The flexural modulus was measured at a test speed of 2 mm/min, and the remainder of the curve was acquired at a test speed of 10 mm/min.

Impact testing was performed in accordance with ISO 179-1eU (2010) [41] on a Zwick PSW B5113.300 with various hammers to achieve 30 to 70% energy absorption. The choice for unnotched samples was based on the knowledge that fiber-filled materials are brittle and show little to no distinction between samples when tested with notched samples.

#### 2.2.5. Heat Deflection Temperature

The heat deflection temperature was tested following ISO 75 method A (1.8 MPa) (2013) [42] on an Instron HV3S filled with silicone oil. After concluding that this is a misrepresentation, behavior testing was also performed with ISO 75 method C (8 MPa) [42] for select samples.

#### 2.2.6. Density

Density measurements were performed in accordance with ISO 1183 method A (immersion method) (2012) [43] on half of the middle part of a tensile sample.

#### 2.2.7. Water Absorption

Water absorption was performed according to ISO 62 (2008) [44]. However, instead of using square plate samples, tensile bar samples were used. Eight samples per material were placed in a one-liter container and then filled with tap water. The water content was measured after 1, 2, 4, 7, 11, 15, 21, and 28 days. After 28 days, half of these samples were tested, and the other half was subjected to two cycles of freezing (−23 °C for 24 h), drying (23 °C at 50% RH for 72 h), soaking (23 °C for 72 h), and finally freezing (−23 °C for 24 h) followed by 28 days to dry before testing.

In parallel, 6 samples were placed in a container containing saturated salt solutions to create relative humidities of 11 and 90%. The samples that were not in direct contact with the solution were subjected to tensile testing after 28 days.

#### 2.2.8. Creep

The standards for creep testing are not very strict; thus, much freedom is allowed. ISO 899-1 (2017) [45] [and ISO 899-2 (2003) [46] specify the equipment and provide pointers to avoid errors. Since the choice of materials and design is generally dictated by either the modulus or strength, the choice was made to set the load to 1/500 of the measured modulus of the samples.

Tension creep was tested on the same equipment used for the static tensile test with the addition of a strain gauge amplifier (HBM QuantumX MX1615B). To compensate for the small temperature drift in the climate-controlled room, creep measurements were performed with two strain gauges attached to the middle point of the samples to inspect slippage between the sample and the clamp [47] (HBM LY1-350-6). A macroextensometer was used as a backup.

The strain gauges were glued onto two samples. The first was mounted on the test sample, and the second was mounted on a second sample suspended near the test sample. The glue used was a methylmethacrylate and a peroxide (the combination is sold as HBM X60). To allow enough processing time, the peroxide was cooled to a temperature below −10 °C. The gel was allowed to cure for 16–24 h to avoid variation (according to the data sheet, only 30 min was required at room temperature).

Creep in flexure at various temperatures (20 °C, 40 °C, and 60 °C) was measured on the same equipment as the HDT. An external camera was used to record the deformation.

#### 2.2.9. Welch’s *t*-Test

To verify that the results are significantly different, a two-sample location test was used to check whether the means were significantly different. Welch’s *t*-tests were used for analysis. A regular (or Student) *t*-test assumes equal variance and sample test size, which is rare (especially considering that the number of test samples varies relatively little).

## 3. Results

The manufactured PP, PS, PLA, and their compounds were visually inspected. Then, the materials were mechanically tested, and their tensile, flexural, impact, heat deflection temperature, water absorption, and creep properties were evaluated. The annealed PLA underwent the same tests, and the influence of the thermal treatment was assessed.

### 3.1. Material Color

A visual inspection was initially conducted on the extruded pellets and their respective injection-molded test specimens. All manufactured systems containing the pellets and the respective injection-molded materials are presented in Appendix A. The pellets exhibit minimal color variation but differ in gloss. A greater gloss indicates that the wood fiber reinforcement is embedded in the polymer matrix, which naturally exhibits this prominent characteristic. Conversely, the addition of reinforcement reduces this gloss due to the inherent visual properties of the wood fiber [48]. This reveals that a greater proportion of wood fiber was exposed at the sample surface, where complete encapsulation by the matrix was less likely [49]. Additionally, the increase in fibers tends to enhance the crystallinity of polymers, consequently rendering them opaque in color [33,50,51,52]. As shown in Figure 2, PLA1+F17.5 exhibited wetting characteristics, while the increase in fiber volume (PLA1+F35 and PLA1+C35) gave the composite a dry appearance. However, using the PLA2 variant with lower viscosity recovers some of that glossy appearance. The contents of the compounds with maleic anhydride and PLA-grafted maleic anhydride are between those of PLA1+F35 and PLA2+F35.

On the other hand, the injection molded bars show a significant variation in color, as is clearly visible in Figure 3. The four pure materials are the same color as the pellets (see Appendix A). The glass fiber has become more grayish and has a marble look. PS+F35 is very dark with white-washed parts and shows multiple specks. PP+F35 is the second-lightest wood compound and shows clear specks. PLA2+F35 is the lightest test bar and shows the fewest specks. The color of the composites is influenced by the visual characteristics of the components, which in turn are affected by the processing parameters [53]. For example, the darker color of PS+F35 is due to the darker hue of the polymer compared to others and the higher processing temperature needed. Above 200 °C, the thermal degradation of the lignocellulosic elements begins, contributing to the darkening of the injection-molded PS samples [34,53].

The five compounds based on PLA1 have clear color differences. Researchers have reported that above a 20% volume load, the addition of wood fibers moderately changed the lightness of plastic wood composites [54]. An increase in the fiber load from 17.5% to 35% was observed. However, coarse fibers have been assessed for better color stability [55]. Temperature measurements of the molten material in the extrusion mold showed a more significant deviation than the barrel sensor’s readings. The fine fibers exhibited the greatest deviations, with PLA1+G-MA+F35 showing a temperature deviation of 7 °C. This is mainly a consequence of shear heating, as the lower viscosity variant did not exhibit this pronounced behavior. Since higher temperatures tend to degrade the fibers, this would explain the darker color of PLA1+F35, as maleic anhydride can mitigate this effect [56].

### 3.2. Static Properties, Heating Deflection Temperature (HDT), and Density

#### 3.2.1. Static Properties of Pure Polymers and Their Compatibilized Compounds

Table 4 shows a summary of the results of the static tests for the pure polymers and their compatibilized compounds. The addition of C320 enhanced the tensile modulus of PP (+327%), PS (+141%), and PLA1 (+142%). The restraint of the chain movement of polymeric matrices by wood reinforcement is the main contributor to increasing the tensile modulus [22,57,58,59]. The tensile strengths of PP (+73%) and PLA1 (+22%) improved, and that of PS decreased (PS: −7%). The same tendency followed for the flexural properties when the flexural modulus increased (PP: +313%; PS: 129%; PLA: 114%), and the flexural strength of PP (+33%) and PLA1 (+9%) increased, but that of PS (−20%) decreased. The better mechanical performance of the PP [60,61] and PLA [25] can be attributed to the formation of covalent linkages between the maleic anhydride and hydroxyl groups of cellulose and also to hydrogen bonds [62].

The addition of a wood fiber reinforcement to PS reportedly reaches a tensile strength saturation between 30% and 40% of the fiber load, corresponding to the 35% used in this study [63]. The research indicates that PS with 40% wood fiber shows increased tensile strength with 0.6 wt.% maleic anhydride, while the composite in this study contains approximately 0.45 wt.% [64,65]. It is important to highlight that the reported increase in the ultimate tensile strength (UTS) by Poletto et al. is not significant compared to neat PS [64]. The loss of this property was merely mitigated in comparison to the composite without the presence of a compatibilizing agent [64,66]. This can be attributed to the not-so-effective stress transfer [64,67]. The effectiveness of the WF reinforcement in the presence of a compatibilizing agent appears to be highly dependent on both the processing conditions and the compatibilizing agent’s content [66,68,69,70]. This underscores the necessity for a more precise formulation to enhance the tensile strength of PS [68]. The glass fiber-filled PP shows substantial improvements in both the tensile and flexural moduli (+857% and +910%, respectively) and major increases in the tensile and flexural strengths (+235% and +223%, respectively).

When examining the flexural properties, the PS-based composite exhibits an increase in flexural modulus, primarily attributed to the stiffness of the wood fibers (WFs), leading to the reduced elongation of the composite. However, its flexural strength decreases even in the presence of a compatibilizing agent. This reduction in flexural strength can be attributed to poor matrix/reinforcement anchoring. Although the deleterious effect of excess SMA in PS-based composites is well-known, it is more likely that in the system studied here, the amount of compatibilizing agent was insufficient to promote greater wettability of the fibers by the matrix [68,69,71,72]. This study aimed to maintain similar levels of MA across the systems analyzed. However, the amount added to the PS composites was lower than the optimal level reported, for instance, by Poletto et al. [69]. Additionally, researchers have reported fiber agglomeration at fiber content levels above 30 wt.%, which can adversely affect the composite’s stress transfer mechanism [66]. These two factors together may act as triggers for the collapse of expected properties, such as flexural strength.

Overall, the impact properties of the fiber compounds are lower than those of the pure polymers, and the PLA1 compound shows the smallest reduction (PP: −90%; PS: −50%; PLA1: −44%). The results corroborate those widely reported in the literature [73]. Several factors, including interfacial adhesion [74], the brittleness of wood fibers [75], fiber content and distribution [76], stress transfer [77], and the role of coupling agents [78] influence the impact strength of composites. In the composites analyzed in this study, the observed reduction in impact strength can be primarily attributed to the inherent brittleness of the fibers. This effect is evident even in the PP composite reinforced with glass fibers (PP+G35), which showed a notable decrease in impact resistance. The rigid nature of the fibers likely induces stress concentrations at the fiber–matrix interface, triggering crack initiation under impact loading [75]. Proportionally, this same observation is valid for natural fibers, causing cracks to form and propagate with lower energy requirements [66]. Furthermore, the research has reported a tendency for WF agglomeration at levels above 30 wt.% for fiber sizes like those used in this study, which can also negatively impact the expected stress transfer mechanism in the composites [66].

While coupling agents are known to improve interfacial adhesion and potentially enhance impact strength, previous studies have indicated that reinforcement levels exceeding 10% often result in reduced impact resistance [78]. The findings of this study support this trend, as the composites contained 35 vol.% reinforcement. At this high level, the brittleness of the fibers becomes the dominant factor, overriding potential benefits from improved interfacial adhesion [76]. This brittleness is further exacerbated at elevated fiber contents, making the composite more prone to fracture during impact loading [75].

A benchmark comparison was conducted between the mechanical properties of the PLA, PP, and PS composites measured in this study and the data reported in the literature for WPCs. Composites were obtained through similar processing techniques with similar reinforcement content and geometry. For PP-based composites, the UTS and impact resistance were consistent with previously reported values for the same matrix. However, the elastic modulus of PP+MAPP+F35 exceeded the average values found in the literature [79,80,81]. In the case of PS-based composites, PS+SMA+F35 demonstrated higher tensile strength, flexural strength, and impact resistance than similar systems in other studies [66,82,83,84]. Among the studies examined, the PLA-based composites exhibited higher tensile strength, tensile modulus, and flexural strength, while the flexural modulus was slightly lower than the average reported values [22,85,86].

Based on the static properties, other matrices offer a significant improvement over PP-based WPCs apart from the impact properties (albeit only a minor difference for PLA). The impact properties of PS can be improved by switching to HIPS [87]. Both issues with PLA can be resolved/improved: an impact modifier can be added to improve impact properties or increase crystallization.

#### 3.2.2. HDT and Density of Pure Polymers and Their Compatibilized Compounds

As observed in Table 4, the HDT of polymer matrices improved by adding wood fibers, demonstrating a restriction in polymer molecular chain mobility [22,88,89]. Specifically, polypropylene (PP) exhibited the highest improvement, with an increase of 62.4 °C, followed by polystyrene (PS) at 15.2 °C and polylactic acid (PLA) at only 3.5 °C. These findings highlight the varying effects of wood fiber reinforcement across different polymer matrices. Although adding maleic anhydride enhances the reinforcing effect of wood fibers, PLA benefited the least from this reinforcement compared to PS and PP. This limited benefit can be attributed to the lower melting temperature of PLA relative to PP, as well as the proximity of the testing temperature to PLA’s glass transition temperature (Tg), making PLA more thermally susceptible within this range [90,91]. Furthermore, MA may exert a plasticizing effect on PLA, reducing its thermal resistance [33]. While wood fibers can act as nucleating agents and potentially increase the crystallinity of PLA, these effects are diminished near the glass transition range [33,92]. For PS, the lower HDT increment is largely influenced by the amorphous nature of the matrix. Despite the improved matrix/fiber interaction in the presence of styrene maleic anhydride (SMA), which may enhance reinforcement, SMA can lower the Tg of PS (approximately 100 °C), increasing the sample’s flexibility [93]. PS undergoes rapid structural transitions as an amorphous polymer when approaching Tg, unlike the more gradual changes observed in semi-crystalline PP, which inherently provides superior thermal stability [94,95,96].

The density of the composites primarily reflects the density of their components, as only a few voids were observed in the matrices. Although polypropylene (PP) is the least dense polymer, adding glass fibers makes it the densest composite, given the higher density of glass fibers compared to wood fibers. When equal volumes of wood fibers are added to all composites, PP exhibits the greatest increase in density, followed by polystyrene (PS) and polylactic acid (PLA). This behavior is expected, as PP composites reinforced with wood fibers demonstrated the highest mass-to-volume ratio of added fibers, as presented in Table 1.

#### 3.2.3. Variations in the Static Properties of PLA

Table 5 presents the results for various PLA-based compounds. A Welch *t*-test was performed on the various PLA compounds since the results are closely grouped together (letters in columns correspond to statistically identical groups).

All the compounds showed a decrease in the impact properties and nearly identical tensile and flexural strengths to those of the original polymers. Adding maleic anhydride to the wood composite slightly improved the impact strength and significantly improved the tensile strength compared to the compound without a compatibilizer.

Overall, the modulus increased with the addition of natural fibers, and the addition of maleic anhydride did not further increase the modulus. The increase in the modulus with increasing fiber reinforcement has been widely reported in the literature [34,97,98]. The research has also described a similar trend in the tensile modulus when MA was added at similar concentrations, showing a tendency for superior features to be achieved by increasing the graft degree [25,99]. Regarding the impact resistance, the addition of fibers reduces the toughness of the PLA matrix, which also leads to a decrease in the impact resistance. In response to this reduction, various solutions have been explored, primarily aimed at maintaining the environmentally friendly characteristics of PLA. Although ester derivatives of bio-based plasticizers have shown significant promise as additives for PLA modification, adding fibers still results in toughness levels below the original matrix [100,101]. For improving this property through the use of elastomers, coupling agents, and plasticizers, the impact modifier must disperse effectively within the matrix and establish strong matrix/fiber connections [101]. Additionally, there is an upper limit to the additive content, beyond which impact resistance is negatively affected [102]. These factors are challenging to balance, especially given the polarity differences between the matrix and the fibers [100,103]. As a result, these constraints continue to limit the application of PLA-WPC in scenarios requiring high toughness, particularly when aiming to avoid the use of fossil-based additives [100,101,102].

Considering the type of wood fiber added to the PLA composites, the mechanical properties remained similar, except for UTS and the flexural modulus. Coarse fibers resulted in higher values for these properties. While an increased fiber content enhanced tensile and flexural moduli, it significantly reduced the impact strength. This behavior can be attributed to the sensitivity of the fibers to the mechanical stresses they experience during processing [104]. Fibers that retain a greater effective length within the composite tend to improve its mechanical properties, as longer fibers are better oriented, enhancing the reinforcement effect [105,106].

However, studies indicate that there is a limit to the effective fiber length that can be sustained, beyond which the benefits diminish [107,108]. Similarly, increasing the reinforcement content restricts the mobility of the polymer chains [89]. The rigid nature of the fibers further contributes to stress concentration, leading to a reduction in the impact strength [109]. This underscores the importance of optimizing both fiber length and content to balance the mechanical performance of the composites [110]. Regarding the type of PLA used, there were minimal changes in the mechanical performance. A trend of higher ultimate tensile strength was observed for PLA1, which was also reflected in its respective composites. This outcome was expected, as PLA1’s lower melt flow rate indicates a higher molecular weight compared to PLA2, which directly influences mechanical properties, such as UTS [111].

One of the objectives of this study was to investigate how the method of introducing the coupling agent into the compounder during the processing of PLA composites would influence their mechanical properties. As illustrated in Figure 1B,C, two different mixing strategies were employed. The results indicate that the sequence in which the components were compounded did not significantly affect the final properties of the composites. This finding suggests that the zones within the compounder barrel where the polymer is molten in the presence of the coupling agent play a critical role in the coupling reaction [112].

In this case, the concentration of maleic anhydride (MA) proved to be the most critical factor, as both compounding processes underwent similar cooling stages at the end of the process [113,114]. However, further studies are necessary to evaluate the extent of grafting in PLA when processed with MA under different compounding conditions.

#### 3.2.4. HDT and Density of PLA and PLA Composites

HDT increased with the addition of natural fiber to PLA. This property depends on the degree of crystallinity of the polymer, which is increased by the presence of natural fibers [115,116,117,118,119,120]. Although studies have shown greater matrix/reinforcement interactions when MA is grafted onto the polymer chains, the effect of increased crystallinity may not be observed in PLA [25]. Thus, the addition of MA did not significantly affect the heat deflection temperature. In addition, the influence of the PLA grade on HDTs was not statistically significant. The difference between PLA1 and PLA2 is negligible, except for the maximum strengths of both pure polymers. This is expected because of the lower molecular weight and higher D-isomer content of PLA2. Fiber size offers minimal differences in properties, with only features more dependent on the aspect ratio, tensile strength, and flexural modulus [34,121] showing statistically significant variations.

The density of the composites did not show any significant difference when the same amount of reinforcement was added. PLA 1 and PLA 2 have the same density and the same tendency. In the added level, maleic anhydride did not modify the densities significantly.

#### 3.2.5. Variations in the Static Properties of Crystallized PLA

To enhance PLA crystallinity, multiple samples were annealed in an oven at 105 °C for two hours [118,122]. The sample selection is partially based on the availability of test samples. The temperature was maintained lower within the crystallization temperature window of PLA (100–140 °C) [118,123,124]. The results are again summarized in Table 6.

The results of the additional crystallization for pure PLA and the compatibilized samples show an overall improvement. The samples without compatibilizer show a mix of some properties that improve while others decline. The moduli improved significantly (apart from the tensile modulus of the coarse fiber variant due to the large standard deviation). The strength of the material improves for the pure polymer as well as the tensile strength of the compatibilized material. Nonetheless, it declines for the noncompatibilized materials. Orue et al. [125] reported that annealing polymer composites with natural fibers could damage the matrix/fiber interface. Cracks next to the PLA and wood fiber interphase could be triggered by the shrinkage of the PLA after the annealing process [126]. However, the presence of a compatibilizer improved the adhesion between the PLA matrix and the wood reinforcement, increasing the tensile strength. It has also been reported that heterogeneous nucleation activity on the surface of cellulose reinforcement, in the presence of a compatibilizer, favors the formation of an anisotropic crystal layer that can effectively transfer load from the matrix to the fibers [125,127,128,129].

Due to the low number of impact tests (four or five) for the fiber-filled oven materials and the large standard deviation, it is not possible to say that the properties improved or declined statistically.

The differences between the two fiber lengths are reduced to only the tensile strength, and the flexural modulus is nearly identical. All other properties are not significantly different between the two materials.

#### 3.2.6. HDT and Density of Crystallized PLA

The HDT improved significantly once the feature depended on thermal annealing [112]. When subjected to the crystallization process, polylactic acid (PLA) increased the glass transition temperature, suggesting an enhanced degree of crystallinity in the samples [130]. The addition of fibers further accentuates this increment in heat deflection temperature, indicating that fibers were a crucial factor in improving the HDT, especially since the presence of maleic anhydride (MA) tended to reduce it slightly. This observation corroborates reports of the plasticizing effect of MA found in the literature and the nucleating agent effect of wood fibers [130,131,132]. On the other hand, the presence of MA grafted onto the PLA matrix improved the HDT performance, suggesting better matrix/WF anchoring. After thermal annealing, PLA composites achieved HDT levels comparable to those of PP composites reinforced with wood fiber and superior to those of PS counterparts. However, PP reinforced with glass fiber still exhibited the highest HDT, demonstrating that the stiffness of the reinforcement remained a critical factor for this enhancement. In PLA composites, the thermal annealing process can also contribute to an increased Tg. Singh et al. report an elevation of Tg associated with increased levels of crystallinity when reinforcement contents exceed 10 wt.% [133]. Nevertheless, more in-depth studies on the crystalline behavior of the matrix under the influence of fiber reinforcement are necessary for our understanding of these correlations.

The densities of the thermally annealed samples were higher than those of the non-annealed ones. This result is consistent with the objective of thermal treatment to increase the degree of crystallinity in the samples [112]. The degree of crystallinity in semi-crystalline polymers is directly correlated with their density [134,135], as crystallized polymers exhibit higher density due to the tighter packing of polymer chains in crystalline regions compared to amorphous regions [136].

### 3.3. Water Absorption

The water absorption of composite materials may cause progressive and irreversible alterations in their mechanical properties, hydrolysis, and degradation properties, and service life for compounding that comprises cellulosic reinforcement [85,137]. The water absorption for PP, PS, PLA, and their compounds was plotted for a period of 28 days (see Figure 4 and Figure 5), and the data are in Appendix A. All these data are the sum of the eight tensile bars weighted. The results displayed in Figure 4 demonstrate the difference in water absorption among various polymers and the lack of saturation in the compounds during the observed period.

PP, PS, and PLA are thermoplastics well known for their hydrophobicity. The polymer absorbs water until it reaches saturation. The slight weight variation after saturation indicates the quality of the sample’s surface finish. Water accumulates in the superficial orifices and is removed by wiping prior to weight. The largest decrease was measured with 0.106 g of PS (which had very rough cutting edges), followed by 0.097 g of PP (with a sleek cutting edge). PS is considered a hydrophobic polymer due to its nonpolar aromatic benzene rings in the structure [138]. However, the presence of these benzene rings imparts a slightly higher polarity to PS, enabling more interactions with water molecules compared to polyolefins like polypropylene [138,139,140]. Considering that most materials are somewhere between smooth and rough edges, a variation between samples of 0.1 g can be considered unimportant. Pure PLA1 absorbs a small amount of water (0.6 g).

The addition of glass fibers to polypropylene generally reduces water absorption. Annandarajah et al. reported that increasing the glass fiber content in composites decreases water absorption rates, primarily due to the hydrophobic nature of polypropylene and the dense packing of glass fibers, which minimizes the free volume available for water ingress [141]. However, moisture can migrate through the fiber–matrix interface, leading to increased absorption in composites with poor interfacial adhesion [142,143]. Combined with the presence of minor surface imperfections, where water can accumulate, this explains the slightly higher water absorption observed in PP+G35 compared to neat polypropylene. With the addition of natural fibers, water absorption increases considerably, and after 28 days, it has not yet reached equilibrium for the natural fiber compounds. Research shows that the use of maleic anhydride (MA) reduces water absorption in natural fiber-reinforced composites compared to those without MA [144]. This reduction is attributed to improved coupling between the fibers and the polymer matrix, which narrows the interfacial gaps between them. Consequently, this minimizes the void volume where water can accumulate. However, this does not diminish the hydrophilic nature of the fibers themselves [140,144]. It is clear that there is a large difference between the various polymers. PP-based compounds performed best, and the amount of PS-based compounds absorbed almost doubled. The PLA1-based compound consumed 238% more water than the PP-based compound. This can be attributed to the fact that the diffusion coefficient of PLA is significantly greater than that of PP [145,146]. Unfortunately, for these properties, no direct comparable data are available, even though ISO 62 provides instructions for calculations. Since the focus of these tests is aimed at the mechanical properties and not on absorption, it would be difficult to calculate these parameters from the tensile bars due to the shape (small difference between thickness and width). In addition, for pure polymers, the precision of the balance was too low.

Figure 5 depicts the water absorption behavior of PLA1, PLA2, and their respective compounds. There is a slight difference (less than 0.1 g) between PLA1 and PLA2 for all observed time intervals. Increasing the fiber content in the composites resulted in more water absorption. Although similar behavior can be observed for composites with and without compatibilizers, PLA-G-MA tended to absorb more water (+0.233 g). One of the primary mechanisms by which maleic anhydride-grafted PLA affects water absorption is the formation of ester bonds between the anhydride groups and the hydroxyl groups on the wood fibers. This chemical interaction improves the compatibility of the fibers within the PLA matrix and modifies the composite’s hydrophilic properties. However, studies have indicated that the presence of maleic anhydride may paradoxically increase water absorption due to the enhanced matrix–fiber interaction, which can inadvertently create additional pathways for moisture ingress [147,148].

An apparent deviation from the close grouping is the PLA2 compound, which absorbs more water. This is most likely because the polymer has a lower viscosity, which may result in either (a) a better ability to penetrate the voids in the wood fiber better or less likely to crush the voids, which results in a reduced amount of polymer around the fibers and thus creates a thinner layer allowing water to migrate faster into and between fibers [149]. Alternatively, (b) lower fiber breakage occurred since the shear stresses developed to process PLA 2 are lower than those for PLA1. The presence of more integral fibers results in longer paths where water can traverse faster [150]. This does, however, require additional analysis.

#### 3.3.1. Influence of Moisture Content on Tensile Properties

##### Tensile Properties of Pure Polymers and Their Compatibilized Compounds Under Various Moisture Conditions

Figure 6 and Figure 7 show the results of tensile tests on the pure polymers and their compatibilized compounds, including the standard deviations (for additional information, see Appendix A). The properties of pure PP and PS are not significantly influenced by water, nor are the properties of PP-based compounds. The introduction of water molecules alters the structure of natural fibers, the polymer matrix, and their interface, leading to effects such as water-induced fiber swelling. This swelling not only modifies the fiber structure, but also damages the fiber–matrix interface and induces cracking in the matrix, ultimately resulting in the degradation of the composite’s mechanical properties [151]. As PP-based compounds’ properties remained close to the PP features, this suggests that polypropylene, rather than wood powder, plays a pivotal role in maintaining strength after water absorption of composite samples [152]. Glass fiber-reinforced PP barely shows any modification in the properties due to its hydrophobic nature.

The properties of PS compounds are influenced by moisture. The tensile strength is barely influenced, but the modulus and deformation at the maximum strength are influenced. The modulus decreases slightly with the increasing water content, and the deformation increases marginally. The SFT cycles slightly reduced the E-modulus.

Only the tensile strength and deformation and the ultimate tensile strength of pure PLA1 are influenced. An increase in water content decreases both. After the SFT, the tensile strength slightly decreased. For the compatibilized PLA1 compound, all properties are influenced by an increase in water content. The tensile modulus and strength are negatively influenced, while the deformation at the maximum tensile strength increases. After the SFTs, the tensile modulus and strength decreased. The tensile modulus is only lower than that of the PS compound once the material has soaked and is greater under other moisture conditions than that of the PS and PP compounds. The tensile strength under all conditions was greater than that of the PS and PP compounds. However, PLA exhibits the greatest proportional decrease in this property with increasing environmental humidity and number of SFTs. Moisture susceptibility is known as the primary driving force for PLA degradation [153]. Absorbed water cleaves the polymer to oligomers and causes solubilization of oligomeric fractions [154]. Under the same environmental conditions, PP [154] and PS [155] degradation are negligible. Additionally, PLA absorbed more water than its counterparts, corroborating the findings of Murayama et al. [156], suggesting that the hydrolysis of ester linkages and a reduction in the degree of hydrogen bonding between the MA groups and the WF result in a deterioration of the mechanical characteristics of WPCs, besides the physical damage caused to fiber cells due to swelling and transmitted to the matrix [152].

##### Tensile Properties of Various PLA Materials Under Various Moisture Conditions

Figure 8 and Figure 9 show the tensile results for the various PLA compounds. The properties of the two pure PLAs are nearly identical; only at 90% RH does PLA2 have a marginally higher tensile strength. Their compounds have identical tensile moduli for 10, 50, and 90% RV, but the tensile strength of PLA2 is lower. The properties of PLA2 soaked and exposed to SFT cycles are lower than those of PLA1.

The only difference between the coarse and finer fibers is that the tensile strength of the coarser fiber is approximately 1 to 2 MPa greater under all exposure conditions. There was no significant difference between the PLA-G-MA+F35 and PLA+MA+DCP+F35 compounds; however, they performed slightly better than those without the compatibilizer. The modulus remains slightly higher at 90% RV as well as after the SFT cycle. The tensile strength remains significantly higher.

The properties of the 17.5 V% compound are mixed. The E-modulus remains between that of the pure polymer and that of the 35 V% compound. The tensile strength is always lower than that of the pure polymer. The tensile strength at 10, 50, and 90% RV is also lower than that at 35 V%, but after soaking and SFTs, the tensile strength is greater. This behavior may be due to the composite’s lower fiber volume, resulting in fewer cracks caused by fiber swelling from water absorption. Consequently, humidity negatively impacts the matrix and fiber bonding [157].

### 3.4. Tensile Creep

The main disadvantage of thermoplastics is their poor dimensional stability when subjected to continuous loading over time. This indicates that the principal drawback of thermoplastics is their low creep resistance, which is a crucial hindrance to their further use in industrial applications [158]. Unfortunately, the tensile creep test has less accuracy than expected. Even when two strain gauges are combined with adjusting for temperature changes, the strain curves are not straight and exhibit significant instability or even negative creep (which is disproved when a macroextensometer is used). Furthermore, during the analysis of the creep test findings, it became obvious that, contrary to an initial test, the procedure was insufficiently accurate to compare outcomes that were highly similar. This could be solved with more measurements, but due to the limited quantity of tensile bars, additional measurements were not possible.

#### 3.4.1. Tensile Creep of Pure Polymers and Their Compatibilized Compounds

A clear trend is visible at 1/500 of the tensile modulus of the individual compounds, which results in approximately a 0.2% initial strain. Polymer creep depends on the crystallinity, molecular weight, and static tensile modulus, which typically provide initial comparative insights into the behavior of polymers and their wood-reinforced composites. The polypropylene shows a significantly larger creep deformation, regardless of the presence of a reinforcement. On the other hand, PS and PLA show no large differences between the various compounds (see Table 7). PP exhibited the lowest tensile modulus among the polymer matrices studied. Additionally, it demonstrated the lowest viscosity, indicating a lower molecular weight [159]. These factors enhance the dimensional instability of the polymer, increasing its susceptibility to creep [160]. The presence of glass fibers, characterized by their high tensile strength and stiffness, significantly enhances the load-bearing capacity of the composite. In addition to superior mechanical properties compared to wood fibers, the longer length of glass fibers further contributes to improved performance [161,162]. Consequently, the superior performance of PP+GF35 was expected. However, when comparing the deformation at the end of the test, WF-reinforced PP exhibited lower deformation. It is important to note that the WFs used in this study are shorter than the glass fibers (GFs). Georgiopoulos et al. highlight that fiber size can significantly influence dispersion within the matrix, with smaller fibers potentially achieving better distribution and reduced creep deformation [163].

The extent of deformation in the pure polymers showed an inverse relationship with their tensile modulus; polymers with lower tensile moduli exhibited greater deformation (PP > PS > PLA). The creep per unit of load is a clear indicator of how much the various materials creep. The PLA1 compound creeps the least, and even pure PLA1 creeps less than the PP compounds. The same holds true for the PS compound. Although fiber addition decreases the creep of PP, in addition to inherent molecular features, compounds can have internal defects that decrease the stress transfer efficiency [164].

The accuracy of various PLA materials is a problem. In addition, only a single PLA2 test succeeded due to strain gauge failure and hydraulic control issues.

#### 3.4.2. Tensile Creep of Crystallized PLA and PLA Compounds

Crystallization of the PLA compounds further increased the modulus of PLA with PLA-G-MA and fine wood fibers (see Table 8). Thus, it can handle a (marginally) higher load with more or less the same creep after crystallization. However, due to the accuracy problem, exact comparisons cannot be made.

Despite this limitation, it is important to note that after annealing, the samples exhibited a tendency to reduce deformation over the testing period compared to their non-annealed counterparts. This can be attributed to a potential increase in the crystallinity of the composites, highlighting the ability of thermal treatment to mitigate deformation in composites [165]. Moreover, this finding highlights the potential of annealing as a means to tailor the internal structure of composites for load-bearing applications.

### 3.5. Flexural Creep at Various Temperatures

#### 3.5.1. Flexural Creep of Pure Polymers and Their Compatibilized Compounds

Similar to tensile creep, it is visible that the PP materials have a higher creep at 1/500 of their moduli at 20 °C, while the other materials are close together (see Table 9 and Figure 10). However, in general, the flexural creep resistance tends to be higher than the tensile creep resistance [163].

At 40 °C, the creep resistance decreases for all compounds to various magnitudes, and the unfilled PLA1 shows a more significant creep than the filled PP compounds. Additionally, the PLA1 compound with natural fibers and compatibilizers exhibited significantly reduced creep resistance, displaying values closer to those of PP-filled materials at the same temperature. Additionally, the PLA1 compound with natural fibers and compatibilizers exhibits significantly reduced creep resistance, with values closer to those of PP-filled materials at the same temperature. For PP+F35, the increase in deformation was also pronounced, suggesting that the matrix plays a key role in creep resistance under these conditions [163]. Creep resistance is highly sensitive to the testing temperature. Furthermore, Bledzki et al. argue that the effect of temperature is so pronounced that it may outweigh any improvements achieved by adding coupling agents [166].

At 60 °C, the situation changes radically, and the PLA-based materials have no remaining strength (the glass transition temperature and the HDT temperature of PLA is approximately 55 °C). The creep resistance of PS and its composite is generally superior to that of PP and reinforced PP, with the reinforcement demonstrating significant efficacy in mitigating deformation at 60 °C.

#### 3.5.2. Flexural Creep of PLA Compounds

For the various PLA compounds, the accuracy is too low at 20 °C. At 40 °C, fiber loading seems to be leading, as well as the compatibilizer, which seems to perform slightly better (see Table 10 and Figure 11). PLA2 deforms more than PLA1, which is expected given that the latter exhibits higher viscosity [164]. In the present study, coarse fibers demonstrated better creep resistance at the evaluated temperatures, contrasting with the findings reported by [163]. At 40 °C, PLA1+G-MA+F35 exhibited a slightly better performance compared to PLA1+MA+DCP+F35. However, it cannot be conclusively determined whether one composite had better matrix–fiber anchorage than the other. The difference in measurements may be attributed to the effect of fiber agglomeration. Although the literature suggests that compatibilizing agents enhance fiber dispersion within the matrix, the occurrence of agglomeration cannot be overlooked [163,167].

#### 3.5.3. Flexural Creep of Crystallized PLA and PLA Compounds

Since the differences between the various PLA compounds at 20 °C were small and the number of available crystallized PLA flexural bars was low, the test was only performed at 40 °C and 60 °C. At 40 °C, annealed, both the unfilled and filled compatibilized PLA materials perform significantly better than their uncrystallized counterparts (under slightly higher loads of 16 and 4%, respectively). Annealing plays a pivotal role in enhancing the crystallinity of polymer composites, thereby improving their resistance to creep deformation. As Kaavessina et al. observed, annealing facilitates the recovery of crystalline structures disrupted during manufacturing processes, such as extrusion or injection molding [168]. This recovery leads to a more compact arrangement of polymer chains, which enhances the overall crystallinity [168]. The resulting increase in crystallinity contributes to a more rigid internal structure, making the composite less prone to creep under sustained loads [169,170,171].

However, the benefits of annealing must be carefully weighed against potential drawbacks. Reis et al. highlighted that the growth of spherulites during annealing can introduce edge defects, which may adversely affect the mechanical properties of the polymer [172]. These findings underscore the importance of precise control over annealing parameters to maximize the benefits of increased crystallinity while minimizing the risk of mechanical integrity loss [172]. Proper optimization of these parameters is essential to ensure that the material maintains its enhanced performance without compromising its structural reliability.

At 60 °C, the unfilled PLA rapidly deforms. The filled PLA deforms faster than the PP material; however, it appears that the rate decreases. Again, it is important to consider that these tests are performed at 1/500 of their moduli and that the test load of the filled PLA bar is 79% greater than that of the filled PP (see Table 11 and Figure 12).

## 4. Conclusions

In this study, wood fiber-reinforced composites based on PP, PS, and PLA, along with their respective compatibilized versions, were produced through reactive extrusion. The mechanical properties and the influence of water absorption were evaluated. Glass fibers were used as a comparison and were found to significantly enhance the tensile strength, flexural strength, and HDT of the PP matrix.

Adding fibers to polymer matrices improved the tensile modulus, tensile strength, flexural modulus, flexural strength, HDT, and density of the studied composites. However, it was detrimental to the impact strength. The PS matrix did not show an improvement in tensile and flexural strength, likely due to an inefficient matrix/reinforcement interaction mechanism. In PLA composites, increasing the fiber content resulted in a general improvement in mechanical properties, except for the flexural strength. Coarse fibers demonstrated a better mechanical performance, particularly in tensile strength, compared to fine fibers, while other properties remained largely unchanged.

Compared to PLA2, PLA1 showed a slightly higher tensile strength, likely due to its higher molecular weight and viscosity. The addition of maleic anhydride did not significantly affect the tensile and flexural moduli but improved tensile and flexural strength, suggesting better coupling between PLA and wood fibers. In the presence of the compatibilizer, the tensile, flexural, and impact strengths of PLA composites exceeded those of PP+G35

Different feeding methods for the compatibilizer during compounding did not affect the performance of the composites. Thermal annealing of PLA composites proved to be a viable method for further improving their mechanical properties and HDT.

PLA composites were more susceptible to degradation in the presence of water, as they exhibited the highest water absorption among the studied materials. Under creep conditions, increasing the temperature further reduced the gap in the deformation between PLA and PP composites. While additional precise testing is required, the findings suggest that PLA composites with this fiber content and annealing techniques exhibit mechanical properties comparable to PP+G35, making them a potential substitute in certain applications.

This substitution could reduce the reliance on fossil-based polyolefins, providing a more sustainable alternative for polymer composite applications.

## Figures and Tables

**Figure 1 biomimetics-10-00046-f001:**
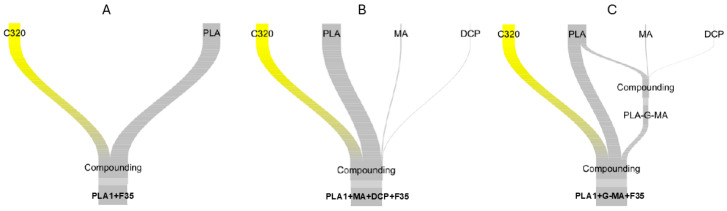
Differences in preparation between (**A**) PLA1+F35, (**B**) PLA1+MA+DCP+F35, and (**C**) PLA1+G-MA+F35. MA and DCP are amplified 10×.

**Figure 2 biomimetics-10-00046-f002:**
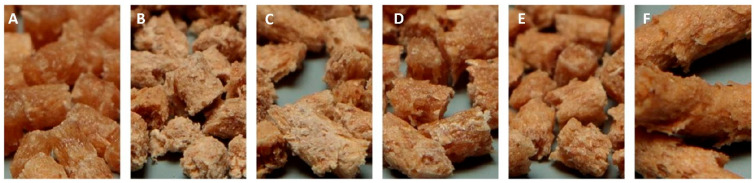
Macro-scale wetting of wood fibers by the polymer matrix. (**A**) PLA1+F17.5; (**B**) PLA1+F35; (**C**) PLA1+C35; (**D**) PLA2+F35; (**E**) PLA1+MA+DCP+F35; (**F**) PLA1+G-MA+F35.

**Figure 3 biomimetics-10-00046-f003:**
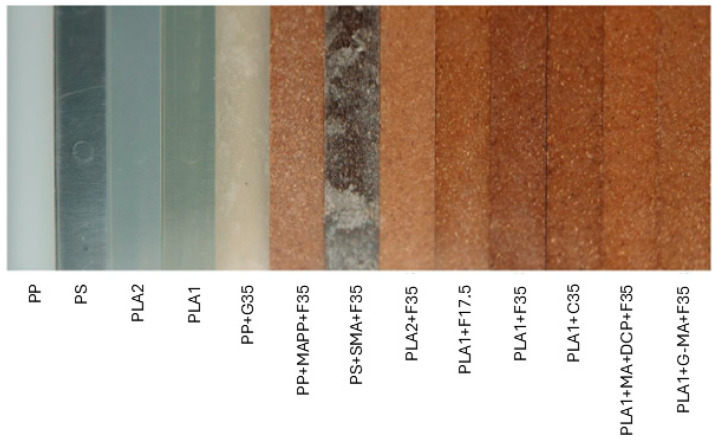
Side-by-side images of all the injection-molded samples.

**Figure 4 biomimetics-10-00046-f004:**
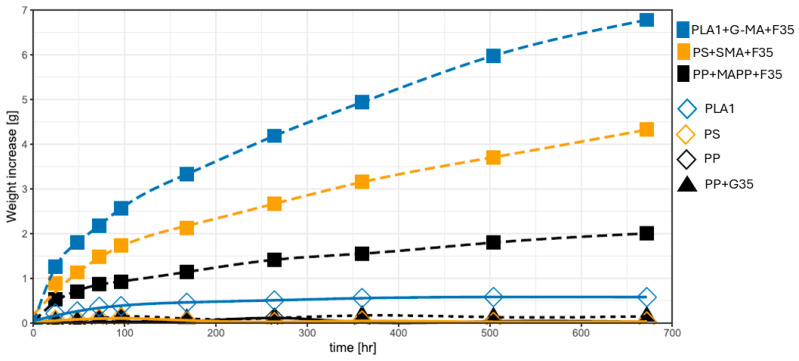
Water absorption behavior of submerged tensile bars fabricated from neat polymers and compatibilized polymer composites.

**Figure 5 biomimetics-10-00046-f005:**
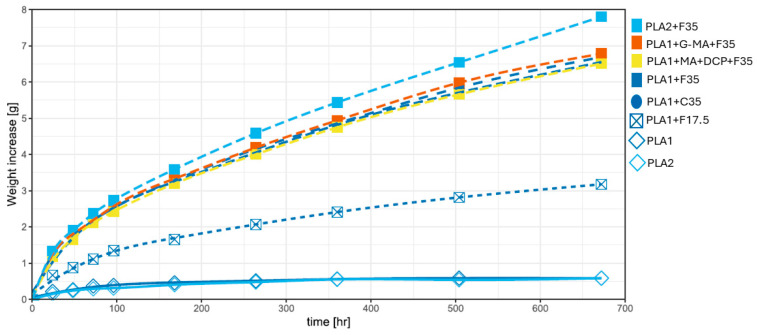
Water absorption of submersed PLA1, PLA2, and their respective composite tensile bars for 28 days. The composites that underwent the test were PLA2+F35, PLA1+G-MA+F35, PLA1+MA+DCP+F35, PLA1+F35, PLA1+C35, and PLA1+F35.

**Figure 6 biomimetics-10-00046-f006:**
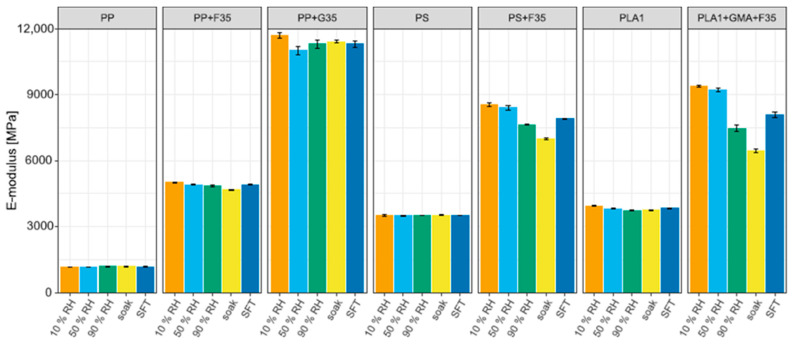
Tensile modulus of pure polymers and compatibilized compounds under various moisture.

**Figure 7 biomimetics-10-00046-f007:**
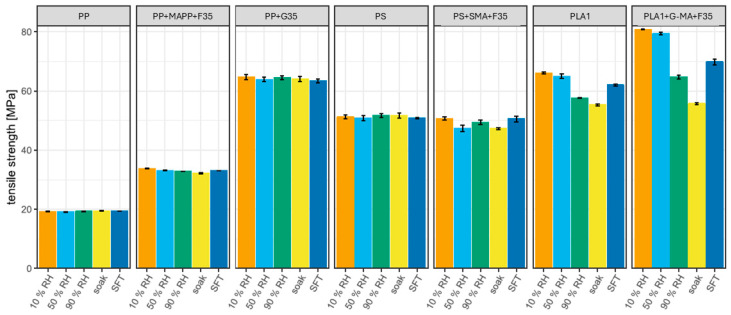
Tensile strength of pure polymers and compatibilized compounds under various moisture conditions.

**Figure 8 biomimetics-10-00046-f008:**
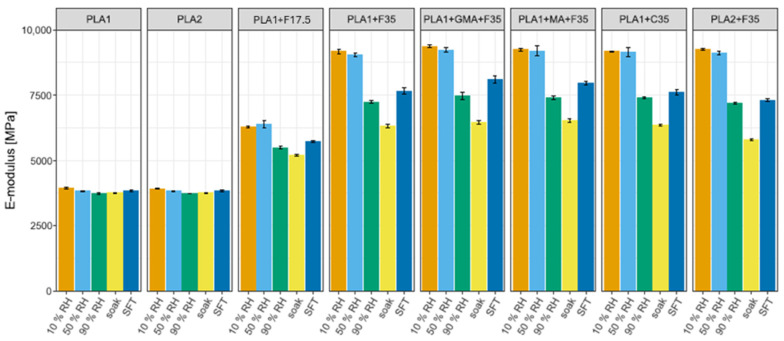
Tensile modulus of pure PLA and PLA compounds under various moisture conditions.

**Figure 9 biomimetics-10-00046-f009:**
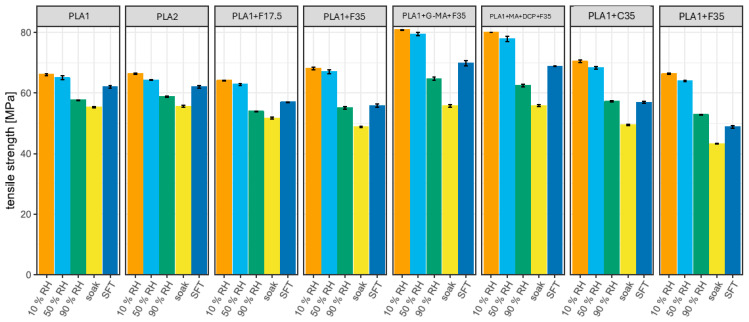
Tensile strength of pure PLA and PLA compounds under various moisture conditions.

**Figure 10 biomimetics-10-00046-f010:**
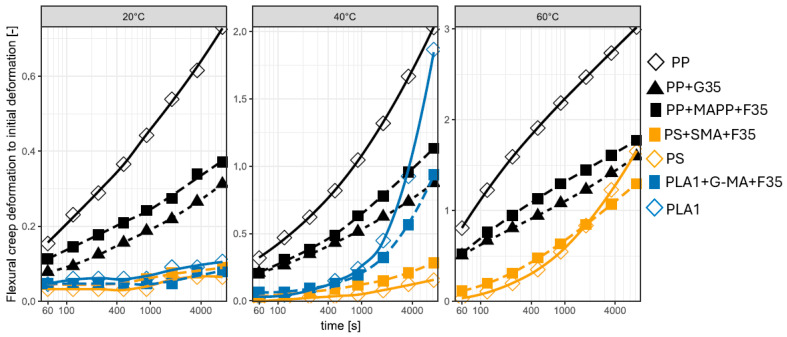
Flexural creep of pure polymers and their compatibilized compounds.

**Figure 11 biomimetics-10-00046-f011:**
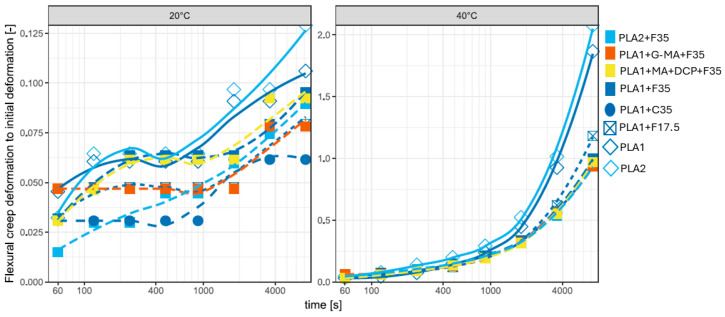
Flexural creep of PLA and PLA-based composites.

**Figure 12 biomimetics-10-00046-f012:**
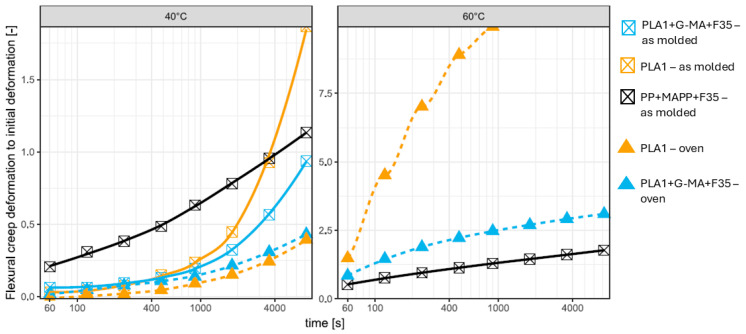
Flexural creep of crystallized PLA and PLA compounds compared to that of the PP composite.

**Table 1 biomimetics-10-00046-t001:** Description of the composites. Samples: F = fine wood fiber (C320), C = course wood fiber (C400), and G = glass fiber (HP3299). Processes: CMP = compounding; IM = injection molding. The material code is as follows: polymer (+additives) (+fiber with fiber content in volume percent).

Material Code	Polymer		Fiber Type/Mass Percentage	Processes
polymer (+additives) (+fiber with fiber content in volume percent)	Polymer	Additives		
PP	Sabic PHC27 (PP)			IM
PP+G35	Sabic PHC27 (PP)		HP3299 [60 M%]	CMP, IM
PP+MAPP+F35	Sabic PHC27 (PP)	MAPP [5 pph]	C320 [47.5 M%]	CMP, IM
PS	BASF 158K (PS)			IM
PS+SMA+F35	BASF 158K (PS)	SMA [5 pph]	C320 [43.5M%]	CMP, IM
PLA1	NW 4043D (PLA)			IM
PLA2	NW 6252D (PLA)			IM
PLA1+F35	NW 4043D (PLA)		C320 [40 M%]	CMP, IM
PLA1+C35	NW 4043D (PLA)		C400 [40 M%]	CMP, IM
PLA1+F17.5	NW 4043D (PLA)		C320 {20 M%]	CMP, IM
PLA1+G-MA+F35	NW 4043D (PLA)	PLA-G-MA [42.9 pph]	C320 [40 M%]	CMP, IM
PLA1+MA+DCP+F35	NW 4043D (PLA)	DCP [0.15 pph]	C320 [40 M%]	CMP, IM
		MA [0.45 pph]		
PLA2+F35	NW 6252D (PLA)		C320 [40 M%]	CMP, IM

**Table 2 biomimetics-10-00046-t002:** Temperature profile of the compounder.

Material	Entry-Zone [°C]	Zone 1 [°C]	Zone 2 [°C]	Zone 3 [°C]	Zone 4–8 [°C]	Die Plate [°C]
PP+G35	45	150	220	220	220	210
PS+SMA+F35	45	150	210	190	185	180
All others	45	150	200	175	175	170

**Table 3 biomimetics-10-00046-t003:** Temperature profile of the injection molding machine.

Material	Zone 1 [°C]	Zone 2 [°C]	Zone 3 [°C]	Zone Nose [°C]	Mold [°C]
PP	180	190	200	190	40
PP+G35	190	210	220	215	40
PS	180	200	220	220	60
PS+SMA+F35	170	185	195	190	40
PLA1, PLA2	170	185	195	195	40
PLA composites	170	180	185	180	40

**Table 4 biomimetics-10-00046-t004:** Static testing and heating deflection temperature (HDT) and density results of the pure polymers and their compatibilized compounds. Appendix A presents the information from the table in a graphical format.

	Tensile			Flexural			Impact	HDT	Density
	Modulus	UTS*	Elongation at UTS	Modulus	UFS*	Elongation at UFS	Unnotched Charpy	Method A	Method A
Compound	MPa	MPa	%	MPa	MPa	%	kJ/m^2^	°C	kg/m^3^
PP	1150 ± 14	19.1 ± 2.62	6.4 ± 0.1	1030 ± 32	31.3 ± 1.1	7.1 ± 0.2	163 ± 2.0	49.3 ± 0.8	901 ± 13
PP+G35	11,000 ± 118	63.9 ± 2.25	1.1 ± 0.1	10,400 ± 403	101 ± 1.74	1.6 ± 0.1	16.6 ± 0.5	146.2 ± 1.4	1438 ± 20
PP+MAPP+F35	4910 ± 60	33.1 ± 2.03	1.8 ± 0.1	4250 ± 247	59.5 ± 1.51	2.8 ± 0.1	15.2 ± 0.4	111.7 ± 1.0	1094 ± 9
PS	3490 ± 42	50.8 ± 2.82	2.1 ± 0.1	3220 ± 106	106 ± 2.16	4.9 ± 0.15	11.5 ± 0.3	83.4 ± 0.3	1049 ± 9
PS+SMA+F35	8400 ± 106	47.4 ± 1.96	0.6 ± 0.05	7360 ± 287	85 ± 1.59	1.2 ± 0.05	5.7 ± 0.2	98.6 ± 0.3	1190 ± 14
PLA1	3820 ± 23	65.0 ± 2.74	2.3 ± 0.1	3430 ± 83	110 ± 2.16	4.6 ± 0.15	19.3 ± 1.6	51.7 ± 0.3	1252 ± 7
PLA1+G-MA+F35	9230 ± 108	79.4 ± 2.05	1.3 ± 0.05	7330 ± 296	120 ± 1.82	1.9 ± 0.1	10.9 ± 0.9	55.2 ± 0.5	1326 ± 9

UTS*—ultimate tensile strength. UFS*—ultimate flexural strength.

**Table 5 biomimetics-10-00046-t005:** Static testing results of PLA and PLA composites. The letters in columns correspond to statistically identical groups. Appendix A presents the same information in a graphical format.

	Tensile			Flexural			Impact	HDT	Density
	Modulus	UTS*	Elongation at UTS	Modulus	UFS*	Elongation at UFS	Unnotched Charpy	Method A	Method A
Compound	MPa	MPa	%	MPa	MPa	%	kJ/m^2^	°C	kg/m^3^
PLA1	3820 ± 23 a	65.0 ± 2.74	2.3 ± 0.1	3430 ± 83 a	110 ± 2.16	4.6 ± 0.1	19.3 ± 0.8 e	51.7 ± 0.3 a	1252 ± 7 a
PLA2	3820 ± 23 a	64.2 ± 1.14	2.3 ± 0.05	3420 ± 79 a	108 ± 1.5	4.7 ± 0.1	19.3 ± 0.8 e	52.2 ± 0.4 a	1252 ± 7 a
PLA1+F35	9040 ± 115 b	67.0 ± 1.2	1.0 ± 0.02	7200 ± 183	99.6 ± 1.2a	1.5 ± 0.1	8.5 ± 0.5 ab	55.2 ± 0.5 b	1324 ± 47 b
PLA1+C35	9150 ± 117 b	68.3 ± 1.2	1.0 ± 0.02	7310 ± 186 b	102 ± 1.2	1.6 ± 0.1	9.2 ± 0.5 bc	55.2 ± 0.5 b	1327 ± 47 b
PLA1+F17.5	6390 ± 82	62.8 ± 1.2	1.5 ± 0.04	5430 ± 138	106 ± 2.0 b	2.5 ± 0.1	13.0 ± 0.6	53.8 ± 0.5	1291 ± 46.1
PLA1+G-MA+F35	9230 ± 118 b	79.4 ± 1.1	1.3 ± 0.03	7330 ± 183 b	120 ± 2.5 b	1.9 ± 0.1	10.9 ± 0.4 d	55.2 ± 0.5 b	1326 ± 47 b
PLA1+MA+DCP+F35	9200 ± 117 b	77.9 ± 1.4	1.3 ± 0.03	7370 ± 188 b	117 ± 2.5 b	1.8 ± 0.1	10.4 ± 0.6 cd	55.1 ± 0.5 b	1326 ± 47 b
PLA2+F35	9110 ± 114 b	63.9 ± 1.1	0.9 ± 0.03	7350 ± 187 b	96.8 ± 1.8	1.5 ± 0.1	7.8 ± 0.5 a	56.7 ± 0.5 b	1324 ± 47 b

UTS*—ultimate tensile strength. UFS*—ultimate flexural strength.

**Table 6 biomimetics-10-00046-t006:** Static testing of crystallized PLA-based composites. Appendix A presents the information from the table in a graphical format.

	Tensile			Flexural			Impact	HDT	Density
	Modulus	UTS*	Elongation at UTS	Modulus	UFS*	Elongation at UFS	Unnotched Charpy	Method A	Method A
Compound	MPa	MPa	%	MPa	MPa	%	kJ/m^2^	°C	kg/m^3^
PLA1	3820 ± 23	65.0 ± 2.74	2.3 ± 0.1	3430 ± 83	110 ± 2.16	4.6 ± 0.1	19.3 ± 0.8	51.7 ± 0.3	1252 ± 7
PLA1 (oven)	4310 ± 34	71.1 ± 1.7	2.1 ± 0.1	3980 ± 174	128 ± 3.4	4.8 ± 0.1	22.7 ± 4.6	60.1 ± 0.6	1259 ± 12
PLA1+F35	9040 ± 115	67.0 ± 1.2	1.0 ± 0.02	7200 ± 183	99.6 ± 1.2	1.5 ± 0.1	8.5 ± 1.1	55.2 ± 0.5	1324 ± 47
PLA1+F35 (oven)	9340 ± 98	65.8 ± 1.7	1.0 ± 0.2	7470 ± 195	95.6 ± 2.6	1.4 ± 0.1	7.4 ± 1.1	118 ± 0.5	1330 ± 35
PLA1+C35	9150 ± 117	68.3 ± 1.2	1.0 ± 0.02	7310 ± 186	102±1.2	1.6 ± 0.1	9.2 ± 0.5	55.2 ± 0.5	1327 ± 47
PLA1+C35 (oven)	9340 ± 78	67.1 ± 1.5	1.0 ± 0.2	7480 ± 152	99.0 ± 3.2	1.4 ± 0.1	9.3 ± 1.4	111.2 ± 0.7	-
PLA1+G-MA+F35	9230 ± 118	79.4 ± 1.4	1.3 ± 0.03	7330 ± 187	117 ± 2.5	1.9 ± 0.1	10.9 ± 0.9	55.2 ± 0.5	1326 ± 47
PLA1+G-MA+F35 (oven)	9530 ± 35	83.6 ± 1	1.2 ± 0.08	7620 ± 164	119 ± 4.1	1.7 ± 0.1	12.9 ± 1.9	117.6 ± 0.3	1327 ± 42

UTS*—ultimate tensile strength. UFS*—ultimate flexural strength.

**Table 7 biomimetics-10-00046-t007:** Tensile creep deformation of pure polymers and their compatibilized compounds.

Material	Tensile Modulus	Load	Initial Deformation	1 h Creep Deformation	Total Deformation	Creep/Initial	Creep/Initial/Load
	MPa	MPa	%	%	%	%	%/MPa
PP	1150 ± 14	2.3	0.235	0.113	0.348 ± 0.020	48.2	20.96
PP+GF	11,000 ± 118	22.0	0.168	0.078	0.246 ± 0.014	46.6	2.12
PP+MAPP+F35	4910 ± 60	9.82	0.173	0.073	0.246 ± 0.08	42.3	4.31
PS	3490 ± 42	6.98	0.179	0.019	0.198 ± 0.012	10.5	1.50
PS+SMA+F35	8400 ± 106	16.80	0.195	0.021	0.216 ± 0.005	10.7	0.64
PLA1	3820 ± 23	7.64	0.199	0.017	0.216 ± 0.011	8.6	1.13
PLA1+G-MA+F35	9230 ± 108	18.46	0.192	0.018	0.210 ± 0.005	9.2	0.50

**Table 8 biomimetics-10-00046-t008:** Tensile creep deformation of various PLA compounds and their crystallized versions that stayed in the oven.

Material	Tensile Modulus	Load	Initial Deformation	1 h Creep Deformation	Total Deformation	Creep/Initial	Creep/Initial/Load
	MPa	MPa	%	%	%	%	%/MPa
PLA1	3820 ± 23	7.64	0.199	0.017	0.216 ± 0.11	8.6	1.13
PLA1 (oven)	4310 ± 34	8.62	0.172	0.020	0.192 ± 0.060	11.4	1.32
PLA1+F35	9040 ± 115	18.08	0.188	0.024	0.212 ± 0.007	12.8	0.71
PLA1+F35 (oven)	9340 ± 98	18.68	0.188	0.018	0.207 ± 0.360	9.8	0.52
PLA1+G-MA+F35	9230 ± 118	18.46	0.192	0.018	0.210 ± 0.008	9.2	0.50
PLA1+G-MA+F35 (oven)	9530 ± 35	19.06	0.182	0.019	0.200 ± 0.420	10.2	0.53

**Table 9 biomimetics-10-00046-t009:** Two-hour flexural creep curves of the pure polymers and their compatibilized compounds. The flexural modulus of the pure polymers and their compounds at different working temperatures is shown in Appendix A.

			20 °C		40 °C		60 °C	
Material	F-mod.	Load	Creep/Initial	Creep/Initial/Load	Creep/initial	Creep/Initial/Load	Creep/Initial	Creep/Initial/Load
	MPa	MPa	%	%/MPa	%	%/MPa	%	%/MPa
PP	1030 ± 32	2.06	73.1 ± 2.3	35.47	203.0 ± 4.6	98.56	302.0 ± 7.4	146.62
PP+GF	10,400 ± 403	20.8	31.3 ± 0.8	1.5	87.5 ± 1.7	4.21	159.8 ± 7.0	7.68
PP+MAPP+F35	4250 ± 247	8.50	37.1 ± 0.6	4.36	113.2 ± 2.3	13.32	177.2 ± 5.2	20.84
PS	3220 ± 106	6.44	6.5 ± 0.2	1.00	15.4 ± 0.5	2.39	165.2 ± 4.3	25.64
PS+SMA+F35	7360 ± 287	14.7	9.1 ± 0.13	0.62	28.4 ± 0.5	1.93	129.6 ± 4.0	8.80
PLA1	3430 ± 83	6.86	10.6 ± 1.4	1.55	186.6 ± 2.3	27.20	-	
PLA1+G-MA+F35	7330 ± 296	14.6	7.8 ± 0.6	0.53	93.5 ± 1.4	6.38	-	

**Table 10 biomimetics-10-00046-t010:** Two-hour flexural creep of PLA compounds. Additional information of flexural creep behavior of PLA and its compounds at different work temperatures is in Appendix A.

			20 °C		40 °C	
Material	F-mod.	Load	Creep/Initial	Creep/Initial/Load	Creep/Initial	Creep/Initial/Load
	MPa	MPa	%	%/MPa	%	%/MPa
PLA1	3430 ± 83	6.86	10.6 ± 1.4	1.55	186.6 ± 2.3	27.20
PLA2	3420 ± 79	6.84	12.9 ± 1.4	1.89	207.7 ± 3.7	30.36
PLA1+F35	7200 ± 183	14.40	9.5 ± 0.5	0.66	100.0 ± 7.2	6.94
PLA1+C35	7310 ± 186	14.62	6.2 ± 0.2	0.42	93.9 ± 5.3	6.43
PLA1+F17.5	5430 ± 138	10.86	7.9 ± 0.6	0.73	118.2 ± 4.4	10.88
PLA1+G-MA+F35	7330 ± 183	14.66	7.8 ± 0.6	0.53	93.5 ± 1.4	6.38
PLA1+MA+DCP+F35	7370 ± 188	14.74	9.2 ± 0.7	0.63	97.0 ± 2.0	6.58
PLA2+F35	7350 ± 187	14.70	9.0 ± 1.2	0.61	94.0 ± 2.4	6.40

**Table 11 biomimetics-10-00046-t011:** Two-hour flexural creep of PP, PLA, and crystallized PLA compounds.

			40 °C		60 °C	
Material	F-mod.	Load	Creep/Initial	Creep/Initial/Load	Creep/Initial	Creep/Initial/Load
	MPa	MPa	%	%/MPa	%	%/MPa
PP+MAPP+F35	4250 ± 247	8.50	113.2 ± 2.3	13.32	177.2 ± 5.2	20.84
PLA1	3430 ± 83	6.86	186.6 ± 2.3	27.20		
PLA1 (oven)	3980 ± 174	7.96	39.4 ± 1.4	4.95	-	-
PLA1+G-MA+F35	7330 ± 152	14.66	93.5 ± 1.4	6.38	-	-
PLA1+G-MA+F35 (oven)	7620 ± 164	15.24	43.1 ± 2.0	2.83	309.7 ± 6.0	20.32

## Data Availability

The data presented in this study are available upon request from the corresponding authors.

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
