# Peer review of "Improving the Long-Term Mechanical Properties of Thermoplastic Short Natural Fiber Compounds by Using Alternative Matrices"

_biomimetics, 2025, doi:10.3390/biomimetics10010046_

Round 1
Reviewer 1 Report
Comments and Suggestions for Authors
General Comment
The manuscript focuses on the effects of polymer types and compatibilizers on properties of wood plastic composites.
Some analysis such as DSC and morphology is required to clarify the discussion.
Consistency of PLA1+G-MA+F35 code needs to be considered in the manuscript.
Properties of PLA1+MA+DCP+F35 composite and PLA1+G-MA+F35 composite are not compared and discussed.
This manuscript needs to be revised. Unclear explanations need to be considered.
1. Introduction
It should be mentioned why PP filled with glass fiber is prepared and compared its properties with WF/PP composites.
2. Materials and Methods
2.1 Materials
MFI of PP and PS should be added.
Why is PLA 6252D chosen for this investigation given its extremely high MFI?
2.2.1 Compounding
It should be proved that PLA-G-MA is prepared. % grafting of PLA-G-MA must be given.
Table 1 should provide a list of all material codes. Table 1 does not display certain material codes, such as PP+F35 and PS+F35.
Are PLA1+MA+F35 (Figure 1) and PLA1+MA+DCP+F35 (Table 1) the same material code?
Does the matrix appear to be PLA-g-MA based on PLA1+MA+DCP+F35? But in PLA1+G-MA+F35, PLA-g-MA serves as a compatibilizer. For what reason are these composite preparations being examined and contrasted?
The fiber content in Table 1 is expressed as a percentage of volume, while the table's unit is M%.
The contents of the glass fiber and wood fiber in PP composites are different. It is unfair to compare the properties of composites.
2.2.4 Static mechanical test
HDT and density are not mechanical properties.
2.2.5 Moisture absorption
It is better to replace moisture absorption with water absorption. Water absorption should be reported in terms of % water absorption.
3. Results and discussion
In every table and graphic, the standard deviation (SD) of the properties must be included.
It is better to use the full name of the mechanical properties in the tables instead of abbreviation or symbol.
Properties of PP+G35 composite should be discussed and compared with other composites since in some sections, no discussion is found.
% crystallinity of polymers and their composites needs to be reported to support the discussion. Moreover, morphology of the composites should be performed to prove adhesion between fiber and polymer matrices.
It should be clarified why adding WF results in increasing HDT of polymers. It is due to increased rigidity or % crystallinity.
The effect of improved adhesion between fiber and matrix on HDT should also be discussed.
3.2 Static properties
Density of polymers and their composites should be discussed.
The effect of wood fiber content and wood fiber grade on PLA composites’ properties needs to be discussed.
It should be clearly explained why impact strength of the composites is lower than polymers.
3.3 Moisture absorption
Water absorption is preferable than moisture absorption. The percentage of water absorption should be used when reporting water absorption.
In the tables, it is preferable to use the complete name of the mechanical properties rather than a symbol or abbreviation.
Since several sections lack discussion, the properties of the PP+G35 composite should be examined and contrasted with those of other composites.
To clarify the discussion, the percentage of crystallinity of polymers and their composites must be reported. Additionally, the composites' morphology should be examined to demonstrate the fiber and polymer matrix' adherence.
The reason why adding WF raises the HDT of polymers needs to be explained. It results from a higher percentage of crystallinity or stiffness.
It is better to present y axis of Figure 4 and 5 as % water absorption.
It is preferable to use sample codes as shown in Table 1 to indicate all samples in Figure 4 and 5.
Glass fiber/PP composite is known by the code GF-PP. GF-PP is not a thermoplastic; it is a composite. Why is the water absorption of GF-PP composite lower than that of wood fiber/PP composite?
Water absorption of PP and PS composites with adding compatibilizer should be reported in Table S2. The influence of compatibilizer on water absorption of the PP and PS composites should be also discussed.
It should be clarified this “Although similar behavior can be observed for composites with and without compatibilizers, PLA-g-MA tended to absorb more water. This was attributed to the highly hydrophilic groups of maleic anhydrides”.
Figure caption of Figure 4 needs to be changed. The samples are polymer composites, not polymer blends. What does it mean under tensile stress conditions.
Figure 5's caption should be changed. It must be specified.
Double-check the sample codes in Figure 6-9. Table 1 does not define a few of them (PP+F35, PS+F35, PLA1+GMA+F35).
Section 3.2 states that the creation of covalent bonds between the cellulose's MA and OH groups improve adhesion in compatibilized composites, which raises the composites' mechanical qualities. Nonetheless, hydrogen bonding between the compatibilizer and wood fiber is mentioned in section 3.3.1.1.
3.4 Tensile creep
It is necessary to talk about PP+GF composite tensile creep.
A 3.5 Flexural Creep Section is preferable.
Table 11's format needs to be reviewed.
To identify every sample in Figures 10, 11, and 12, it is best to utilize the sample codes listed in Table 1.
The impact of fiber and crystallization on the samples' flexural creep is discussed.
This “Since the………………………………PLA flexural bars was slow, the test was ………………….performed at 40 and 60C” needs to rechecked.
4. Conclusions
It is necessary to draw conclusions about how fiber type, fiber content, and adhesion enhancement affect the composites' properties.
Reviewer 2 Report
Comments and Suggestions for Authors
1. Several tests, such as impact resistance and creep, acknowledge limited sample sizes, reducing the statistical reliability of findings.
2. Although PLA grades (PLA1, PLA2) are analyzed, the discussion lacks depth on how viscosity differences translate into specific application advantages.
3. Include aging tests under UV exposure or varying environmental pollutants for a broader understanding of composite durability.
4. Some figures (e.g., moisture absorption, tensile properties under moisture conditions) are densely packed and could benefit from clearer labeling or separation for readability.
5. The tensile and flexural creep sections report issues with test accuracy and repeatability, weakening the conclusions drawn.
6. The study mentions PLA's lower impact resistance but does not explore adding impact modifiers as a potential solution. Investigate the effect of impact modifiers on PLA-based composites for comprehensive mechanical property optimization.
7. While the paper compares WPCs with glass-reinforced composites, it lacks clear benchmarks against industry standards or competing materials. Include a benchmarking section with performance metrics of conventional materials used in similar applications.
8. The paper discusses the environmental relevance of biocomposites but does not quantify life-cycle benefits or economic feasibility. Add a life-cycle analysis (LCA) or cost-benefit discussion to strengthen the argument for industrial adoption.
Round 2
Reviewer 1 Report
Comments and Suggestions for Authors
The consistency of PLA-G-MA needs to be considered in the manuscript. PLA-g-MA, PLA-g-MAH and GMA are utilized.
2. Materials and Methods
2.1 Materials
MFR of PS needs to be double checked. This “MFR 3.7 g/10 min at 20C and 5kg” is incorrect.
What does crystallization ratio mean?
2.2.1 Compounding
Table 1 should provide a list of all material codes. Table 1 does not display certain material codes, such as PP+F35 and PS+F35. Figure 3 shows images of PP+F35 and PS+F35 samples.
From Figure 1 (b), material code as shown in diagram (PLA1-MA+F35) and caption (PLA1+MA+DCP+F35) are different.
3. Results and discussion
The standard deviation (SD) needs to be provided with the average data in Table 4-11.
Maleic anhydride (MA) is not a compatibilizer. It is a monomer. Polymer grafted maleic anhydride (PLA-g-MA, MAPP, SMA) serves as compatibilizer in polymer composites.
It should be clarified why PS+SMA+F35 composite exhibited lower UTS, elongation at UTS, UFS, elongation at UFS, and impact strength than PS.
Water absorption of PP and PS composites with adding compatibilizer should be reported in Table S2.
What does it mean under tensile stress conditions as shown in caption of Figure 4?
4. Conclusions
From "The PS matrix did not exhibit an improvement in tensile and flexural strength, likely due to fiber agglomeration", this conclusion is not consistent with the discussion section. Furthermore, the composites' morphology is not investigated in this experiment.
